# Effect of Spectral Signal-to-Noise Ratio on Resolution Enhancement at Surface Plasmon Resonance

**DOI:** 10.3390/s21020641

**Published:** 2021-01-18

**Authors:** Long Ma, Guo Xia, Shiqun Jin, Lihao Bai, Jiangtao Wang, Qiaoqin Chen, Xiaobo Cai

**Affiliations:** 1School of Instrument Science and Opto-Electronics Engineering, Hefei University of Technology, Hefei 230009, China; malong@mail.hfut.edu.cn (L.M.); bailihao@mail.hfut.edu.cn (L.B.); jiangtaowang@mail.hfut.edu.cn (J.W.); qqchen@mail.hfut.edu.cn (Q.C.); caixiaobo@mail.hfut.edu.cn (X.C.); 2Academy of Opto-Electric Technology, Hefei University of Technology, Hefei 230009, China; shq_king@163.com; 3Special Display and Imaging Technology Innovation Center of Anhui Province, School of Instrument Science and Opto-Electronics Engineering, Hefei University of Technology, Hefei 230009, China; 4National Engineering Laboratory of Special Display Technology, School of Instrument Science and Opto-Electronics Engineering, Hefei University of Technology, Hefei 230009, China

**Keywords:** sensor, refractive index resolution, surface plasmon resonance, spectral signal-to-noise ratio

## Abstract

Refractive index resolution is an important indicator for a wavelength interrogation surface plasmon resonance sensor, which can be affected by signal-to-noise ratio. This paper investigates the impact of spectral signal-to-noise ratio on a surface plasmon resonance sensor. The effects of different spectral powers and noises are compared and verified through simulation and experiments. The results indicate that the optimal resonance wavelength is changed and the refractive index resolution can even be nearly twice as good when the spectral signal-to-noise ratio is increased. The optimal resonance wavelength can be found by changing the spectral power distribution or noise.

## 1. Introduction

Surface plasmon resonance (SPR) has been widely used in drug screening [1,2], biochemical safety [3], food safety [4], environmental monitoring [5], utilization of solar energy [6,7,8], and other fields [9,10] in the last twenty years. SPR, which is an optical phenomenon characterized by excitation between the noble metal and the dielectric layer, has many merits, such as label-free real-time [11] monitoring, high sensitivity, and low cost. Meanwhile, the sensor can detect the sample regardless of whether the state of the sample is gas [12], liquid [13], or solid [14]. According to the interrogation method, the SPR system can be divided into angle interrogation [15], phase interrogation [16], amplitude interrogation [17], and wavelength interrogation [18]. The resolution of the wavelength-modulated SPR is changed with the wavelength. Therefore, finding the optimal resonance wavelength is an important basis to determine the working spectral band. However, the spectral signal-to-noise ratio (SNR) can affect the determination of the optimal resonance wavelength.

Refractive index (RI) resolution has always been an important performance index. RI resolution is used to evaluate the smallest change of the resonance wavelength that can be distinguished in the SPR structure [19,20,21]. The RI resolution is defined as the ratio of the sensor’s detection accuracy to sensitivity [22,23]. When the noise increases, the resonance wavelength fluctuates greatly. The changes of the spectral power distribution (SPD) can cause an asymmetrical change in the full width at half maximum (FWHM) of the SPR curve, and the asymmetry generates peak finding errors and increases the fluctuation of the resonance wavelength. Therefore, the spectral SNR [24] curve is a key factor affecting sensor performance. Vlek et al. changed the sensor layer structure and the metal layer structure to enhance the resolution of the SPR sensor [25]. Maharana et al. changed the prism material and metal layer material to enhance the resolution of the SPR sensor [15]. However, the methods, which improved the performance by changing the structure of the sensor, may increase the difficulty of production. Improving the resolution of the sensor without changing the structure and production of the sensor is a key research direction in the traditional Kretschmann structure. Chen et al. analyzed the influence of the resonance wavelength on the resolution of the sensor [22]. As a precursor of this paper, Zhou et al. analyzed the influence of three different SPDs on sensor resolution [26]. On the basis of these articles, this paper analyzes the influence of spectral SNR.

In this paper, the influence of spectral SNR on SPR is further studied based on the traditional Kretschmann structure. The SPD and noise are adjusted to obtain the optimal resonance wavelength, so that the working spectral band is then determined. In Section 2, we simulate the influence of the spectral SNR on the sensitivity, detection accuracy, and resolution by mathematical modeling. The influence of the spectral SNR is further obtained through physical experiments. In Section 3, the results of the simulation and the experiments are analyzed. Through the spectral SNR analysis, the optimal working spectral range of different types of sensors is obtained. Moreover, the influence of the spectral SNR on the best resolution and the best resonance wavelength is able to be evaluated.

## 2. Materials and Methods

### 2.1. Transfer Matrix Modeling

According to Maxwell’s equations and boundary conditions [26,27], the reflectivity of the *N*-layer structure can be calculated by the transmission matrix method:(1)$$M=\prod_{k=2}^{N-1}M_{k}=\left(\begin{array}{cc} M_{11} & M_{12}\\ M_{21} & M_{22} \end{array}\right)=\left(\begin{array}{cc} \cos\beta_{k} & \frac{\left(-i\sin\beta_{k}\right)}{q_{k}}\\ \left(-iq_{k}\sin\beta_{k}\right) & \cos\beta_{k} \end{array}\right),$$
where $\beta_{k}=\left(\frac{2\pi d_{k}}{\lambda }\right)\times {\left(\varepsilon_{k}-n_{1}{}^{2}\sin\theta_{1}\right)}^{\frac{1}{2}}$, $q_{k}=\frac{{\left(\varepsilon_{k}{}^{2}-n_{1}{}^{2}\sin\theta_{1}\right)}^{\frac{1}{2}}}{\varepsilon_{k}}$, *d_k_* represents the *k_th_* layer thickness, *n_k_* represents the RI in the *k_th_* layer, and *θ*_1_ represents the incident angle.
(2)$$R_{p}={\left(\phi_{p}\right)}^{2}={\left[\frac{\left(M_{11}+M_{12}q_{N}\right)q_{1}-\left(M_{21}+M_{22}q_{N}\right)}{\left(M_{11}+M_{12}q_{N}\right)q_{1}+\left(M_{21}+M_{22}q_{N}\right)}\right]}^{2},$$
where *R_p_* is the reflectance. The incident angles *θ*_1_ are 42.6°, 42.7°, 42.75°, 42.8°, 43°, 43.3°, 43.5°, 43.7°, 44°, and 44.6°. The parameter values in the mathematical model will be listed below.

### 2.2. Model Parameters

In Figure 1, *n_BK_*_7_ can be calculated as follows [26,28]:(3)$$n_{BK7}=(\frac{1.03961212\lambda^{2}}{\lambda^{2}-0.00600069867}+\frac{0.231792344\lambda^{2}}{\lambda^{2}-0.0200179144}+\frac{1.03961212\lambda^{2}}{\lambda^{2}-103.560653}{)}^{\frac{1}{2}},$$

According to the Drude–Lorentz model, the thickness of the gold film is 50 nm and the dielectric constant can be calculated as follows [26]:(4)$$\varepsilon_{m}=\left[1-\frac{\lambda_{c}\lambda^{2}}{\lambda_{p}{}^{2}(\lambda_{c}+i\lambda )}\right],$$
where *λ_p_* = 168.26 nm is the plasma wavelength and *λ_c_* = 8934.2 nm is the collision wavelength.

The three kinds of SPD are simulated in the sensor simulation.

As shown in Figure 2, among the three SPDs, SPD1 is the ideal light source, and SPD2 and SPD3 change the slope of the SPD to obtain different changing trends of spectral SNR and performance indicators.

In the actual measurements, the uncontrollable noise mainly comes from the detector, which is mainly divided into the readout noise, dark noise, fixed mode noise, and photoelectric noise [22,29]. Each noise is independent, and the total noise can be expressed as follows:(5)$$N=\sqrt{N_{R}{}^{2}+N_{D}{}^{2}+N_{P}{}^{2}+N_{F}{}^{2}},$$

As shown in Figure 3, the noise simulation value is given separately, and the final calculated simulation value *N* is given. *N_R_* is dependent on the circuit design of the instrument and is mainly generated when the analog signal is transformed into a digital signal. *N_R_* can be calculated as follows [29]:


(6)$$N_{R}=std(S_{bias}),$$


*N_D_* is the dark noise in the detector because of the thermal movement of particles, which produces current at the output end. *N_D_* can be calculated as follows [29]:(7)$$N_{D}=std(S_{D}),$$

*N_F_* is the fixed mode noise caused by the difference in pixel dark current, which is mainly determined by the manufacturing process. *N_F_* can be calculated as follows [29]:(8)$$N_{F}=S_{F},$$

*N_p_* is the photoelectron noise, which is determined by the statistical difference of particle arrival at the detector. *N_p_* can be calculated as follows [29]:(9)$$N_{P}=\frac{\sqrt{n_{e-}}}{g},$$
where *S_bias_* is a bias signal, *S_D_* is the dark current signal and wavelength point at the current wavelength point, *n_e−_* is the number of detected photoelectrons, and *g* is gain constant. Given the influence of total noise, the SPR measurement curve is expressed as follows [26]:(10)$$S_{m}=R_{p}\times SPD+N,$$
where *S_m_* is the measurement curve, *R_p_* is the reflectance curve, and *N* is the total noise. This paper qualitatively analyses the effects of spectral SNR through the spectral SPR curve. During the simulation, the results are changed by adjusting the noise and SPD. The RI resolution is used as an evaluation index to judge the influence of the spectral SNR.

Three different SPDs are constructed and two noise levels are added. The standard deviation of resonance wavelength is regarded as noise for each resonance wavelength. The light intensity value is used as the signal value in the current wavelength. The spectral SNR corresponding to the resonance wavelength is simulated, as is shown in Figure 4a,d,g. The spectral SNR is calculated as follows:(11)$$SNR=10\times \log_{10}(abs(\frac{P_{s}}{P_{n}})),$$

*P_s_* is the spectral power signal, and *P_n_* represents the resonant wavelength noise.

Figure 4 shows the spectral SNRs and fitted SPR curves under three spectral powers along with two types of noise. Spectral SNRs show the same trend as SPD, and the fitted SPR curves of the spectral power under different wavelengths are given. The fitted SPR curves are changed under different spectral SNRs. In the next part, we analyze the changes in sensor performance based on the changes in the fitted curves.

### 2.3. RI Resolution

The resolution formula of RI can be obtained as follows [30]:(12)$$\delta_{n}=\frac{\delta_{\lambda }}{S_{n}},$$
where $\delta_{n}$, $\delta_{\lambda }$, and $S_{n}$ are the RI resolution, RI detection accuracy, and RI sensitivity of the SPR sensor, respectively [31]. *S_n_* is an important indicator of the sensor’s static characteristics and is defined as the ratio of the output change to the input change. In the SPR sensor, when the RI of the sample changes, the resonance wavelength *λ* also changes. The RI sensitivity can be calculated as follows:(13)$$S_{n}=\frac{\Delta \lambda }{\Delta n},$$

The RI of the sample is from 1 to 1 + ∆*n* (∆*n* = 0.00001), the deviation of resonance wavelength is obtained, and the sensitivity of the sensor is calculated and fitted by Equation (13). The fitted results are shown in Figure 5.

As shown in Figure 5, the sensitivity increases with resonance wavelength. The trend between resonance wavelength and sensor sensitivity under different conditions is the same. The increase in sensitivity with wavelength is consistent with the results obtained by mathematical model analysis [30]. There is a slight difference in sensitivity, which is mainly due to the influence of the spectral SNR on the wavelength accuracy in the peak seeking process. The difference in the SPD results in the shift of the resonance wavelength, thus the sensitivity is affected.

$\delta_{\lambda }$ is mainly affected by the FWHM and minimum value of the SPR curve. To verify this result, the effects of three trending spectral SNRs are simulated. The resonance wavelength corresponding to each angle is simulated 1000 times, and the standard deviation of the resonance wavelength here is used as the detection accuracy of the current wavelength.

As displayed in Figure 6a,b, the growth rate in the resonance wavelength of 500–650 nm is small, indicating that the resonance wavelength value is relatively stable. When the resonance wavelength is 650–800 nm, as the resonance wavelength increases, the detection accuracy growth rate increases violently. Because the spectral SNR is very small, the fluctuation of the detected resonance wavelength changes violently, which leads to an increase in the standard deviation of the resonance wavelength and an obvious decrease in system performance. The results show that the spectral SNR can affect the detection accuracy of the current resonance wavelength.

In the simulation, three spectral power curves and two noise levels are presented. The noise and SPD are combined to obtain six simulation results. The influence of the spectral SNR is obtained through methods of controlling variables comparison.

#### 2.3.1. Same Noise Levels at Different SPDs

In this case, the changing trend of resolution is obtained by simulating the three SPDs under the same noise level. Figure 7 shows the effect of noise on RI resolution.

As shown in Figure 7, when the resonance wavelength is 500–650 nm, the resolution of the system changes monotonically. When the resonance wavelength is 650–800 nm, the resolution of the system presents different trends in two cases. The reason for the phenomena is that the spectral SNR of the system decreases rapidly, which means the noise will cause more impacts on the system. The impacts are so severe that the system cannot maintain its own modulated mode, resulting in a different trend of the resolution changes. Thus, it is important to find the optimal resonance wavelength. The optimal resonance wavelength is obtained by changing the SPD.

#### 2.3.2. Same SPD at Different Noise Levels

When the SPD is kept constant, the influence of noise on the sensor resolution can be observed. The influence of different noise levels on the current resonance wavelength and the best resolution is given in Figure 8. The effect of the different noise levels at the same SPD is obtained.

### 2.4. Experimental

The system consists of a light source (halogen lamp, Daheng Optoelectronic Technology Co., Ltd., beijing, China), a collimator (Φ5.0 mm, SMA905, Daheng Optoelectronic Technology Co., Ltd., beijing, China), a polarizer (Φ25.4 mm, FU-PZP-Y24, Daheng Optoelectronic Technology Co., Ltd., beijing, China), a SPR device (right-angle prism BK7, Daheng Optoelectronic Technology Co., Ltd., beijing, China, gold film thickness 50 nm), and a spectrometer (USB4000+ Ocean Optics, Weihai Optical Instrument Co., Ltd., Shanghai, China). The spectral curve of the halogen tungsten lamp is unstable for a short period, which affects the accuracy of the sensor experiment. The measurement experiment can be carried out when the measured spectral curve does not show a large jump. The light source is warmed up for 5 min and the measurement begins when the light source is stable. The measurement needs to be stopped after a period of measurement to prevent the stability of the light source from deteriorating. In the next part, the experimental results are described.

Figure 9a shows the spectral curves of different integration times from the same spectrometer. The fitted spectral SNR curve for the spectral curves of different integration times is shown in Figure 9b. The measured SPR curves of the two spectral signals after adjusting the integration time in the SPR device are shown in Figure 9c,d.

## 3. Results and Discussion

### 3.1. Simulation Results

As shown in Table 1, when the noise level doubles, the spectral SNR of the same SPD decreases, and the optimal resonance wavelength decreases. When the signal value is SPD3, the optimal resonance wavelength increases with the noise level. There are two reasons for this situation. On the one hand, due to the limitation of the measurement range, the maximum resonance wavelength of the measurement is around 800 nm (as shown in Figure 4). On the other hand, the light intensity is weak, and the optimal resonance wavelength is around 800 nm, which is the change caused by noise (as shown in Figure 8c). As the spectral SNR decreases, the numerical of the resolution increases, which means a decrease in resolution.

### 3.2. Experimental Results

In the experiment, the RI of the sample was changed at equal intervals and changes in the spectral curve were recorded 1000 times. Based on the data obtained from the experimental operations above, the sensitivity and detection accuracy of the current resonance wavelength was calculated, and the current resonance wavelength resolution was modeled in Equation (12), as shown in Figure 10.

### 3.3. Discussion

The spectral SNR was changed under different light source wavelengths to determine the performance index of the SPR device. The optimal resonance wavelength was found, and the measurement was taken near the optimal resonance wavelength position to improve the reliability and accuracy of the results. The relationship between spectral SNR and RI resolution between replacement device components and different types of devices may be different from the results of this experiment, but the overall trend should be the same. This experiment provides a technical reference for the performance evaluation of different systems of wavelength-interrogated SPR.

## 4. Conclusions

The spectral SNR obviously affects the optimal resonance wavelength and the optimal resolution. In the experiment, the optimal resonance wavelength and the optimal resolution were obtained at two integration times. On the spectral power curve, the effect of the spectral SNR on the resonance wavelength could be obtained through experiments. The influence in the change of integration time on the spectral SNR is shown in Figure 9b. With the extension of the integration time (T1–T2), the spectral SNR increases, and the optimal resonance wavelength of the sensor increases. Meanwhile, the numerical resolution decreases, which means an increase in resolution. The practicality of surface plasmon resonance devices are greatly improved because the spectral SNR is a common physical quantity for different coupling and sensing layer structures. The replacement of components in the system may cause changes in the measurement results. Therefore, each replacement of components requires recalibration of the SPR equipment.

## Figures and Tables

**Figure 1 sensors-21-00641-f001:**
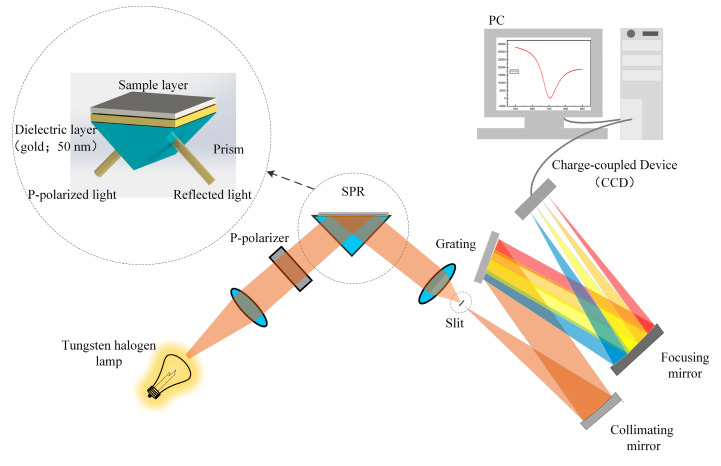
Schematic of a surface plasmon resonance (SPR) sensor.

**Figure 2 sensors-21-00641-f002:**
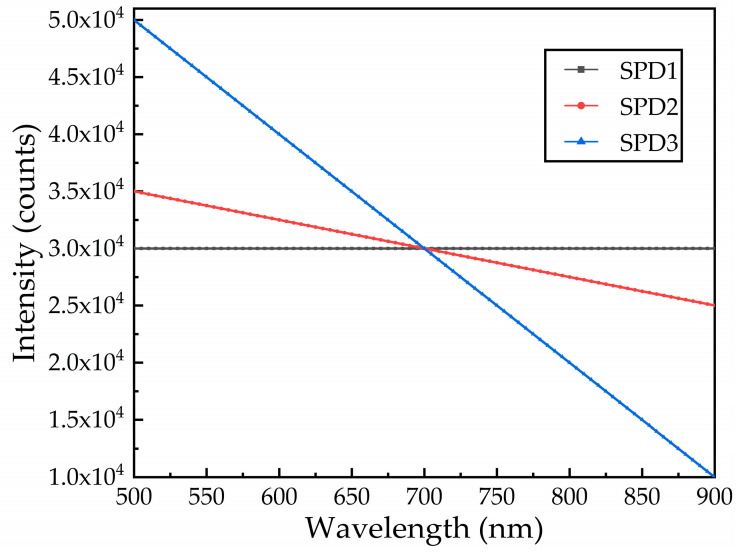
Simulate three spectral power distribution (SPD) curves [26].

**Figure 3 sensors-21-00641-f003:**
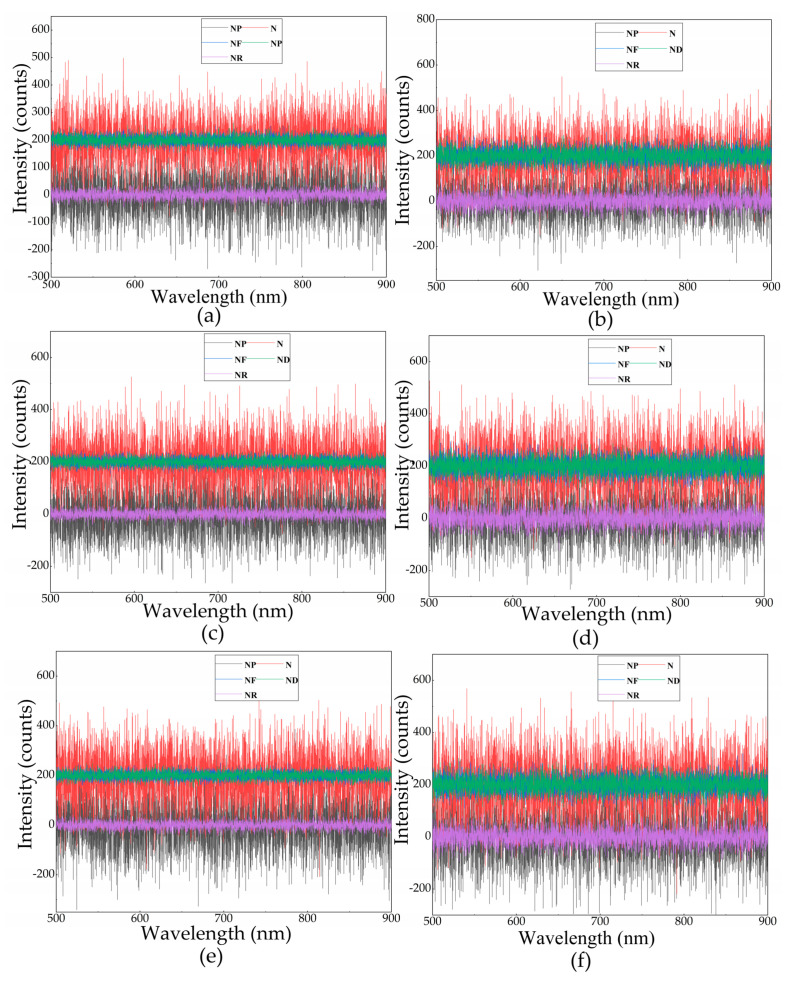
Noise value in simulation: (**a**) noise level (under SPD1), (**b**) double noise level (under SPD1), (**c**) noise level (under SPD2), (**d**) double noise level (under SPD2), (**e**) noise level (under SPD3), and (**f**) double noise level (under SPD3). *N* is the final calculated simulation value; *N_p_* is the photoelectron noise; *N_F_* is the fixed mode noise caused by the difference in pixel dark current; *N_D_* is the dark noise in the detector; and *N_R_* is dependent on the circuit design of the instrument.

**Figure 4 sensors-21-00641-f004:**
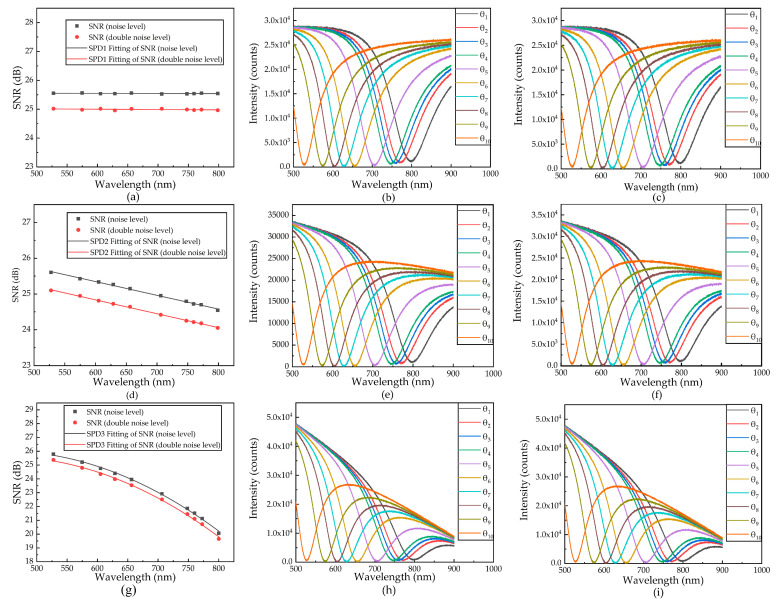
(**a**) Simulation of the spectral SNR fitting curve under two noise levels and SPD1. (**b**) Fitted SPR curve with a double noise level (under SPD1). (**c**) Fitted SPR curve with a noise level (under SPD1). (**d**) Simulation of the spectral SNR fitting curve under two noise levels and SPD2. (**e**) Fitted SPR curve with a double noise level (under SPD2). (**f**) Fitted SPR curve with a noise level (under SPD2). (**g**) Simulation of the spectral SNR fitting curve under two noise levels and SPD3. (**h**) Fitted SPR curve with a double noise level (under SPD3). (**i**) Fitted SPR curve with a noise level (under SPD3).

**Figure 5 sensors-21-00641-f005:**
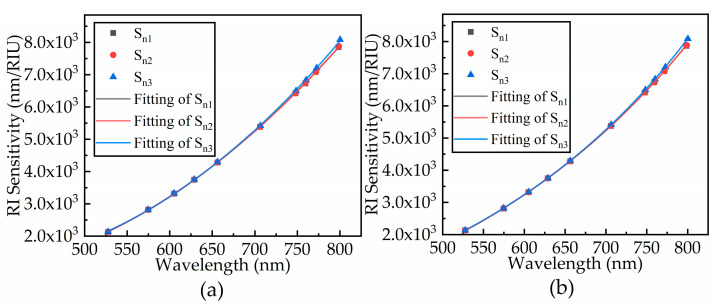
Three signal refractive index (RI) sensitivity curves for (**a**) noise level and (**b**) double noise level.

**Figure 6 sensors-21-00641-f006:**
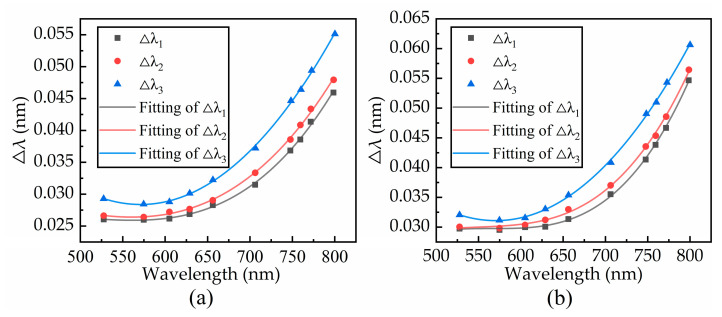
Three signal standard deviation curves for (**a**) noise level and (**b**) double noise level.

**Figure 7 sensors-21-00641-f007:**
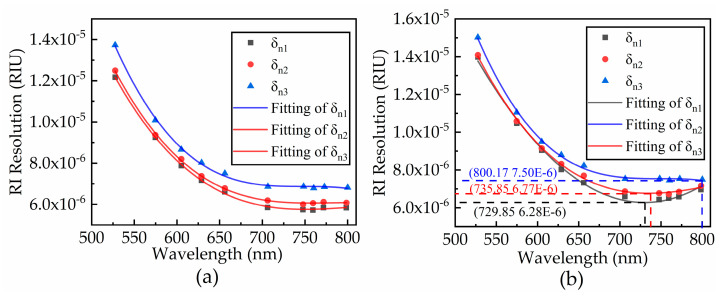
Three signal RI resolution curves for (**a**) noise level and (**b**) double noise level.

**Figure 8 sensors-21-00641-f008:**
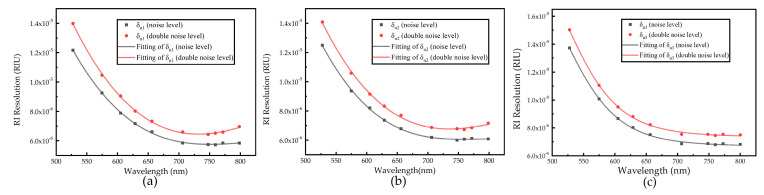
Resolution fitting curve of two noise levels (**a**) under SPD1; (**b**) under SPD2; and (**c**) under SPD3.

**Figure 9 sensors-21-00641-f009:**
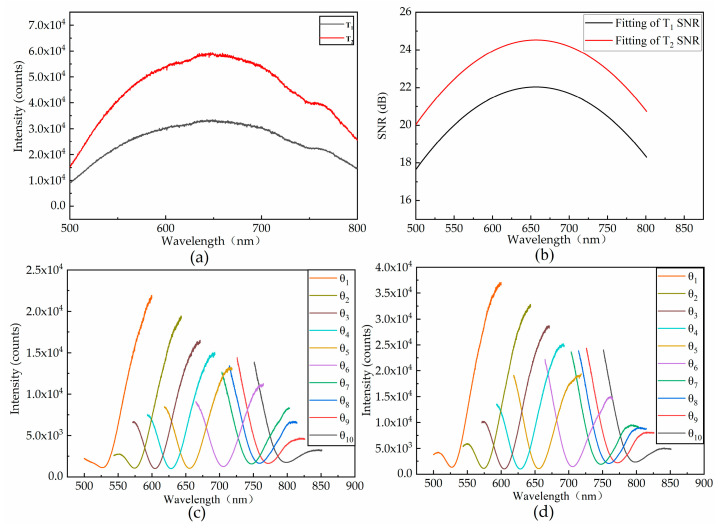
(**a**) SPD curves received at two different integration times. (**b**) Fitting of the spectral SNR curve under two different integration times: (**c**) integration time T1 (SPR curve); (**d**) integration time T2 (SPR curve).

**Figure 10 sensors-21-00641-f010:**
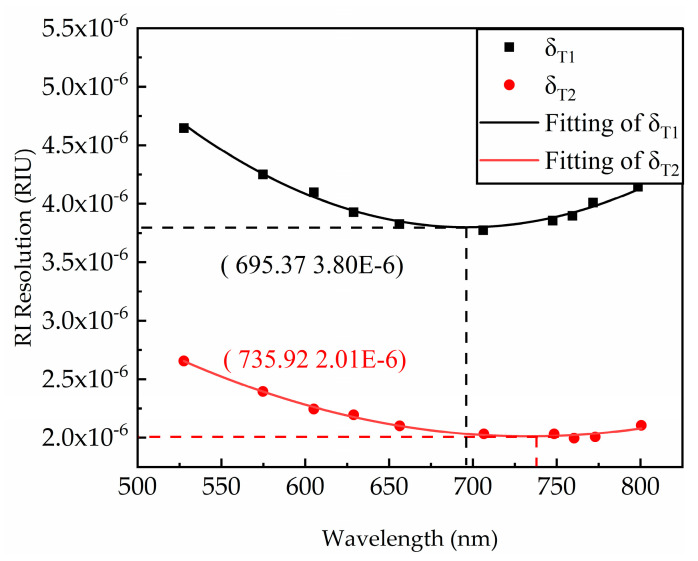
The optimal resonance wavelength at the integration time T1 (integration time 8 ms) is 695.37 nm, and the best RI resolution is ${3.80\text{×}10}^{-6}$ RIU. When the integration time is T2 (integration time 15 ms), the optimal resonance wavelength is 735.92 nm, and the best RI resolution is ${2.01\text{×}10}^{-6}$ RIU.

**Table 1 sensors-21-00641-t001:** SPR resolution simulation results.

	Spectral Power Distribution	Optimal Resonance Wavelength (nm)	Best RI Resolution (RIU)
Noise level	SPD1	759.30	5.72 × 10^−6^
	SPD2	795.15	6.08 × 10^−6^
	SPD3	798.23	6.78 × 10^−6^
Double Noise level	SPD1	729.85	6.28 × 10^−6^
	SPD2	735.85	6.77 × 10^−6^
	SPD3	800.17	7.50 × 10^−6^

## Data Availability

Not applicable.

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
