# Peer review of "Effect of Spectral Signal-to-Noise Ratio on Resolution Enhancement at Surface Plasmon Resonance"

_sensors, 2021, doi:10.3390/s21020641_

Round 1

Reviewer 1 Report

In this manuscript, Ma et al. simulated a method to improve the resolution by changing the combination of spectral signal-to-noise ratio in different bands. The band can be effectively selected and the resolution is nearly doubled as the signal-to-noise ratio changes. This manuscript is well organized, I recommend it published in sensors after addressing the following questions.

1.   In the manuscript, SPD is a variable that should be expressed in italics, but the method of expression is not uniform, such as line 50.

2. In Fig.3(f), the legend of the image is not modified according to the format.

3.    “Ps is the spectral power signal value, and Pn is the resonant wavelength noise value” is recommended to modify to “Ps is the spectral power signal, and Pn represents the resonant wavelength noise”.

4.      Are there duplicates in the simulation results in the two parts of the manuscript 2.3.1 and 2.3.2? If there is no repetition, it should be clearly stated in the physical meaning of the expression.

5.      There are too few references in this paper, and the author needs to supplement more than ten latest and relevant references.

Author Response

Manuscript ID: sensors-1063347 Type: Communications

Title: Effect of Spectral Signal-to-Noise Ratio on Resolution Enhancement at Surface Plasmon Resonance

Author: Long Ma, Guo Xia, Shiqun Jin, Lihao Bai, Jiangtao Wang, Qiaoqin Chen, and Xiaobo Cai;

Dear Reviewer:

Thank you for your valuable comments on our manuscript. We have revised this article according to your comments and suggestions. The main corrections in the manuscript and responses to the reviewer’s comments are as follows:

Responds to the reviewer’s comments:

  1. Response to comment: In the manuscript, SPDis a variable that should be expressed in italics, but the method of expression is not uniform, such as line 50.

Response: From this question, we know that the reviewers are very careful about this manuscript. To show the significance of the manuscript, we modified a description of the background in the first part. Therefore, the first appearance of the proper term (spectral power distribution) has been adjusted to Page 1 line 39. At the same time, we have also made sentence adjustment in terms of suggested by the reviewers.

  1. Response to comment:In Fig.3(f), the legend of the image is not modified according to the format.

Response: I'm very sorry that our negligence in the article cause the inconsistent style of the legend in Fig.3(f). We have modified the legend in Fig.3(f) and reimported the modified picture.

  1. Response to comment: Psis the spectral power signal value, and Pn is the resonant wavelength noise value” is recommended to modify to “Ps is the spectral power signal, and Pn represents the resonant wavelength noise”.

Response: Thank you very much to the reviewers for the modification of the grammar in the manuscript. We have modified the grammar in this sentence to make the presentation of the article clearer in Page 5 line 119.

  1. Response to comment: Are there duplicates in the simulation results in the two parts of the manuscript 2.3.1 and 2.3.2? If there is no repetition, it should be clearly stated in the physical meaning of the expression.

Response: In the manuscript, 2.3.1 and 2.3.2 may not be reused. We are pleased to emphasize the significance of these two parts. In the analysis of the influence of the distribution of noise and spectral power on the SPR sensor, we use the method of controlling variables. Therefore, the methods of controlling the noise level and spectral power distribution are used in the two parts to obtain the optimal resonance wavelength and the optimal resolution.

In 2.3.1, we change the spectral power to obtain the optimal resonance wavelength and the optimal resolution. Then, we change the noise in 2.3.2 to get the best resonance wavelength and best resolution changes at the same spectral power.

  1. Response to comment: There are too few references in this paper, and the author needs to supplement more than ten latest and relevant references.

   Response: We are very grateful to the Reviewers for suggesting the manuscript. We do have deficiencies in the research background investigation in page.1 line 24-34. For this reason, we once again explored the hot directions of SPR research and added them to the references of this manuscript in page.11 line 260-271 and page.12 line 288-396. The background part of the paper and the modifications to the references are shown below.We have made a lot of changes to the error you raised. We are benefit a lot from your reviews,thank you very much! 

Sincerely,

Guo Xia

Reviewer 2 Report

This manuscript seeks to draw conclusions regarding the connection between spectral signal-to-noise ratio (SNR) and the functional properties of a surface plasmon resonance (SPR) sensor system, with particular regard being paid to differing noise levels. Whilst SPR modelling experiments examining the impact of SNR are useful to sensor scientists, I feel that this manuscript has a number of major flaws. The biggest issues are:

  1. The content of the manuscript overlaps with that of a previous publication (reference 17) to a large degree. This overlap of content is concerning for two reasons, firstly because it erodes much of the novelty of this current paper, reducing it to an incremental study adding minimal additional insight into the topic, and secondly because the authors fail to make the significance of the content of reference 17 clear to the readers in comparison to the content in the current paper. Reference 17 has many of the equations given in the current manuscript, one figure that is identical and a number of figures that are quite similar, yet it is only cited once with very little context given.
  2. The authors have not adequately defined what the new insights provided by the paper are. Whilst some potentially interesting data are presented, it is not clear that this really draws any conclusions that aren't either already known/established or are fairly obvious. The usefulness of the conclusions and the associated analysis needs to be made much clearer.
  3. I am concerned that some of the statements made in the manuscript are inconsistent with the data presented, are discussed with insufficient detail or are unclear. More specific details on this are provided below.

Some further specific comments are detailed below. All these points would also require changes to the manuscript text.

  1. The manuscript has numerous grammatical errors that need to be corrected (too many to list).
  2. Page 1 line 40 ". . . and greatly the RI." This phrase is not clear and needs to be reworded.
  3. Page 2 line 45 ". . . on the traditional A structure . . ." What do you mean by "A structure" in this context? Clarify your meaning here.
  4. Page 6 lines 167-168 "After the resonance . . . 650-800 nm." Are you meaning to say that you are referring to the range of 650-800 nm rather than, as your current wording suggests, wavelengths above the 650-800 nm range? Clarify this in the manuscript.
  5. The heading for section 2.3.2 makes no sense and needs to be rewritten.
  6. Figure 8. This optical arrangement is fairly conventional and so an image of it is not required.
  7. Page 7 line 188 ". . . after a period of measurement . . ." How is this time period objectively determined?
  8. Page 8 lines 197-198 "The fitted spectral . . . in Figure 8b." Don't you really mean figure 9b? Correct this in the manuscript.
  9. Page 8 lines 203-204 "As shown in . . . resolution also increase." This comment is not consistent with the data in Table 1. Spectral SNR is expected to decrease with increasing noise, so for this comment to be true then all values for optimal resonance wavelength and best RI resolution should decrease with the decrease in SNR when the noise is doubled. Instead, from Table 1 it is seen that one of the optimal resonance wavelengths and all of the best RI resolutions in fact increase with decrease in SNR at given spectral power distributions. The apparent contradictions need to be clearly resolved and reasons for the observed phenomena clearly given.
  10. Page 9 line 219 "Different effects are . . . of the devices." This statement is vague and lacks detail. You should make it completely clear what effects you are referring to here and how they may differ between devices.
  11. Page 9 line 227 ". . . is shown in Fig.8(b)." Once again this appears to be an incorrect reference to the figure.
  12. Page 9 lines 227-229 "As the integration . . . spectral SNR curves." Whilst increasing the incubation time is resulting in an increase in the optimal resonance wavelength, it is also associated with a decrease in the resolution (as expressed in RU). Whilst this decrease in the resolution figure is an improvement, it is still a decrease in the resolution metric (as expressed in RU) and not an increase.
  13. Figure 2. This figure is identical with one given in reference 17 and so a citation to reference 17 needs to be given in the Figure 2 legend.
  14. Can you succinctly explain in the manuscript the rationale for the selection of the three spectral power distributions used in this paper?
  15. Page 7 lines 176-178. "Different rate noise . . . given in Figure 7." This very brief description does not clarify what the actual amount of noise was that was introduced into the simulation and that was later doubled in the analysis. Quantification of the noise factor and the rationale for its use in the model should be made clear in the manuscript.

Author Response

Manuscript ID: sensors-1063347 Type: Communications

Title: Effect of Spectral Signal-to-Noise Ratio on Resolution Enhancement at Surface Plasmon Resonance

Author: Long Ma, Guo Xia, Shiqun Jin, Lihao Bai, Jiangtao Wang, Qiaoqin Chen, and Xiaobo Cai;

Dear Reviewer:

Thank you for your contributions to our manuscript, and thanks to the reviewer for their careful reading and excellent comments concerning our manuscript entitled “Effect of Spectral Signal-to-Noise Ratio on Resolution Enhancement at Surface Plasmon Resonance” (Manuscript ID: sensors-1063347). We have studied comments carefully and have made a correction which we hope meet with approval. The main corrections in the manuscript and responses to the reviewer’s comments are as follows:

Responds to the reviewer’s comments:

  1. Response to comment: The content of the manuscript overlaps with that of a previous publication (reference 17) to a large degree. This overlap of content is concerning for two reasons, firstly because it erodes much of the novelty of this current paper, reducing it to an incremental study adding minimal additional insight into the topic, and secondly because the authors fail to make the significance of the content of reference 17 clear to the readers in comparison to the content in the current paper. Reference 17 has many of the equations given in the current manuscript, one figure that is identical and a number of figures that are quite similar, yet it is only cited once with very little context given.

Response: Part of the content in the manuscript is referenced from reference 17. Because these two papers come from the same research group. The previous publication (Reference 17) only considered the influence of the input signal (SPD) on the SPR resolution. Based on the precursory research, we consider again. The influence of noise on sensor resolution and optimal resonance wavelength is discussed. The result of this research is a staged progress. Secondly, the simulation algorithm uses the same basis (transfer matrix model), so the parameter settings and models are the same. In the manuscript, we added references to the parts used in previous articles. Thank you very much for your careful reading of our current manuscript and previous publications of this research group. There are indeed deficiencies in the citation of the literature of this research group.

We marked the citation positions of references [26] and [29] in the manuscript.

  1. Response to comment: The authors have not adequately defined what the new insights provided by the paper are. Whilst some potentially interesting data are presented, it is not clear that this really draws any conclusions that aren't either already known/established or are fairly obvious. The usefulness of the conclusions and the associated analysis needs to be made much clearer.

Response: We are very glad to highlight the original contribution in the manuscript, in contrast to current research on sensor performance. The research ideas and innovation of the paper are as follows:

Innovation:

In the manuscript, the sensor coupling structure and the change of the metal layer can increase the difficulty of manufacturing the sensor. The improvement of resolution based on the traditional structure can increase the practicality of the sensor.

Key information in this research of this manuscript: Based on the traditional Krestchmann structure, the influence of the spectral SNR on the best resolution and the best the best resonance wavelength position is judged by changing the system's spectral SNR. The practicability of SPR devices is greatly improved, because the spectral SNR is a common physical quantity for different coupling and sensing layer structures. An effective method for analyzing the best performance of surface plasmon resonance system based on spectral SNR is proposed. By changing the SPD and noise, the SPD is adjusted to obtain the best resolution position so that the best resolution position can exist in the working range.

Data interpretation (Data logic):

         It is really true as reviewer suggested that there are two logical problems with the interpretation of the data in the manuscript. The main logic problem is that when we double the noise level, the signal-to-noise ratio is reduced. The logic used in the manuscript is the situation that the signal-to-noise ratio increases. Secondly, as the numerical of the resolution increases, this indicates that the resolution is decreasing.

Research ideas:

[1] Maharana P K, Jha R. Chalcogenide prism and graphene multilayer based surface plasmon resonance affinity biosensor for high performance[J]. Sensors and Actuators B: Chemical, 2012, 169: 161-166.

Key information in this reference: In this study, the performance of the SPR sensor was enhanced by changing the coupling structure (chalcogenide prism) and the metal layer structure (the number of graphene layers).

Innovation: Changing the number of different graphene layers under different refractive index (RI) conditions improves the detection performance parameters of the sensor.

[2] Vlek J , Jaromír Pitora, Michal Lesňák. Design of Plasmonic-Waveguiding Structures for Sensor Applications[J]. Nanomaterials, 2019, 9(9).

Key information in this reference: Knowing that the number of graphene layers can enhance the performance of the sensor, different coupling methods (waveguide coupling) are used to analyze the relationship between the coupling method and the multilayer parameters.

Innovation: The performance of the sensor is enhanced by changing the metal layer structure and the coupling method of the sensor.

[3] Chen Z , Liu L , He Y , et al. Resolution enhancement of surface plasmon resonance sensors with spectral interrogation: resonant wavelength considerations[J]. Appl Opt, 2016, 55(4):884-891.

Key information in this reference: In the study of the SPR wavelength modulation type Krestchmann system, the performance indicators of different resonance wavelength positions are obtained through simulation to obtain different resolutions. The resolution of the current sensor is improved by selecting different incident angles to obtain different resonance wavelengths and selecting the working interval.

Innovation: A reference index is given for the resolution of the wavelength modulation sensor at different resonance wavelength positions

[4] Zhou C , Xia G , Wang G , et al. Effect of Spectral Power Distribution on the Resolution Enhancement in Surface Plasmon Resonance[J]. Photonic Sensors, 2018, 8(4). (Our research group)

Key information in this reference: In the study of SPR wavelength modulation Krestchmann system, different resonance wavelength positions correspond to different resonance wavelength performance changes, but in addition to the effect of resonance wavelength, the change of the light source can also affect the change of the sensor system shape and further affect the new performance of the system.

Innovation: In the sensor wavelength system, the change of spectral power distribution is added to the performance change of different resonance wavelength positions, and the influence of different spectral power on the current system performance change is obtained.

  1. Response to comment: I am concerned that some of the statements made in the manuscript are inconsistent with the data presented, are discussed with insufficient detail or are unclear. More specific details on this are provided below.

Detailed modification

  • Response: The manuscript has numerous grammatical errors that need to be corrected (too many to list).

Sorry for the grammatical error in the manuscript. With the joint efforts of all authors, we re-organized the grammar of the manuscript. We hope that the modification of the grammar in the manuscript can get your approval. If you find that there are still grammatical problems in the manuscript, please contact us. Thank you very much.

  • Response: Page 1 line 40 ". . . and greatly the RI." This phrase is not clear and needs to be reworded.

Page 1 line 39-41: The meaning of the sentence emphasizes the influence of the spectral power distribution on the SPR curve. Our previous expression was not clear enough, so we changed the sentence to ‘The change of the spectral power distribution (SPD) can cause an asymmetrical change in the full width at half maximum (FWHM) of the SPR curve, and the asymmetry generates peak finding errors and increases the fluctuation of the resonance wavelength.’

  • Response: Page 2 line 46 ". . . on the traditional A structure . . ." What do you mean by "A structure" in this context? Clarify your meaning here.

Page 2 line 49-50: Thanks the reviewers for reading carefully. This is indeed our negligence in revising the manuscript. The expression has been changed to ‘. . . on the traditional Kretschmann structure . . .’.

  • Response: Page 6 lines 167-168 "After the resonance . . . 650-800 nm." Are you meaning to say that you are referring to the range of 650-800 nm rather than, as your current wording suggests, wavelengths above the 650-800 nm range? Clarify this in the manuscript.

Page 7 line 160-163: Regarding the segmentation description of the image, the statement of the limited range is wrong, and we have revised it in accordance with the reviewer’s comments. This sentence was modified to ‘As displayed in Figs.6 (a) and (b), the growth rate in the resonance wavelength of 500-650 nm is small, indicating that the resonance wavelength value is relatively stable. When the resonance wavelength is 650-800 nm, as the resonance wavelength increases, the detection accuracy growth rate increases violently.’.

  • Response: The heading for section 2.3.2 makes no sense and needs to be rewritten.

Page 7 line 167 and Page 8 line 179: The signal value used in the manuscript is SPD, so we replaced the signal in the title with SPD. Title 2.3.1 and 2.3.2 was rewritten as ‘Same noise levels at different SPD.’ and ‘Same SPD at different noise levels.’.

  • Response: Figure 8. This optical arrangement is fairly conventional and so an image of it is not required.

Line 214-218(Original manuscript): We appreciate it very much for this good and sweet suggestion, and we have done it according to your ideas.

  • Response: Page 7 line 188 ". . . after a period of measurement . . ." How is this time period objectively determined?

Page 7 line 189-191: The halogen light source has a large amount of jitter in the spectral curve at the beginning of the measurement, which can affect the final result of the measurement because the jump of the spectral curve may eventually act on the uncertainty of the sensor. The determination of the waiting time mainly depends on observing the changes of the current spectral power curve. ‘The spectral curve of the halogen tungsten lamp is unstable for a short period, which affects the accuracy of the sensor experiment. The measurement experiment can be carried out when the measured spectral curve does not show a large jump.’ is added to this part of the manuscript.

  • Response: Page 8 lines 197-198 "The fitted spectral . . . in Figure 8b." Don't you really mean figure 9b? Correct this in the manuscript.

Page 9 lines 199-200: Sorry it is our mistake. The picture number at this position has been modified.

  • Response: Page 8 lines 203-204 "As shown in . . . resolution also increase." This comment is not consistent with the data in Table 1. Spectral SNR is expected to decrease with increasing noise, so for this comment to be true then all values for optimal resonance wavelength and best RI resolution should decrease with the decrease in SNR when the noise is doubled. Instead, from Table 1 it is seen that one of the optimal resonance wavelengths and all of the best RI resolutions in fact increase with decrease in SNR at given spectral power distributions. The apparent contradictions need to be clearly resolved and reasons for the observed phenomena clearly given.

Page 8 lines 205-211: It is really as reviewer suggested that there are two logical problems with the interpretation of the data in the manuscript. The main logic problem is that when we double the noise level, the signal-to-noise ratio is reduced. The logic used in the manuscript is the situation that the spectral SNR increases. Secondly, as the numerical of the resolution increases, which meanings that the resolution is decreasing.

When the noise doubles, the spectral SNR decreases and the numerical of the resolution increases, which means that the resolution decreases. However, it appears that the optimal resonance wavelength becomes larger as the noise increases in the case of SPD3. There are two reasons for this situation. On the one hand, due to the limitation of the measurement range, the maximum resonance wavelength of the measurement is around 800nm (as shown in Fig.4). On the other hand, the light intensity is weak, and the optimal resonance wavelength is around 800nm, which is caused by noise (as shown in Fig.8(c)). In the case of SPD3, the best wavelength position is the maximum point of the current wavelength. The shift of the wavelength under the two noise levels is due to the effect of noise on the resonance wavelength.

  • Response: Page 9 line 219 "Different effects are . . . of the devices." This statement is vague and lacks detail. You should make it completely clear what effects you are referring to here and how they may differ between devices.

Page 10 line 226-228: In the manuscript, changes in the experimental structure and experimental methods have a greater impact on the resolution of the sensor. The innovation of the manuscript is that for different sensor structures and experimental methods, the test equipment can have an impact on the sensor resolution, but the sensor resolution trend can remain consistent. The sentence was changed to ‘The relationship between spectral SNR and RI resolution between replacement device components and different types of devices may be different from the results of this experiment, but the overall trend should be the same.’.

  • Response: Page 9 line 227 ". . . is shown in Fig.8(b)." Once again this appears to be an incorrect reference to the figure.

Page 9 line 235: Thank you very much for reading carefully, the picture number at this position has been modified.

  • Response: Page 9 lines 227-229 "As the integration . . . spectral SNR curves." Whilst increasing the incubation time is resulting in an increase in the optimal resonance wavelength, it is also associated with a decrease in the resolution (as expressed in RU). Whilst this decrease in the resolution figure is an improvement, it is still a decrease in the resolution metric (as expressed in RU) and not an increase.

Page 10 lines 235-237: I'm very sorry. There is also an error in the logical expression (same as question 9). The logic of this sentence has been modified to ‘With the extension of the integration time (T1-T2), the spectral SNR increases, and the optimal resonance wavelength of the sensor increases. Meanwhile, the numerical of the resolution decrease, which means the increases in resolution.’.

  • Response: Figure 2. This figure is identical with one given in reference 17 and so a citation to reference 17 needs to be given in the Figure 2 legend.

Page 9 line 241: The reference [26] to the SPD curves has been given.

  • Response: Can you succinctly explain in the manuscript the rationale for the selection of the three spectral power distributions used in this paper?

Page 5 line 82-83: When selecting the simulation curve, first of all we considered the impact of the ideal light source, so we chose SPD1, followed by the spectral power curve with changes in the current spectral SNR, and the rate of change was also a research direction, so we chose three types.

  • Response: Page 7 lines 176-178. "Different rate noise . . . given in Figure 7." This very brief description does not clarify what the actual amount of noise was that was introduced into the simulation and that was later doubled in the analysis. Quantification of the noise factor and the rationale for its use in the model should be made clear in the manuscript.

Page 8 line 183-185: I’m very sorry. It is indeed our problem that we want to express the effect of different noise levels.

Page 3-4 line 84-107: In order to facilitate the reproduction of the article, a clear definition is given for the physical quantities used in the simulation. In the manuscript, the current noise level of each noise is given in the data model for SPR curve.

The gold film is widely used in the SPR system, so in the simulation process we use the gold film as the metal layer of the classic Krestchmann structure, and give the parameters used in the simulation. Our research group has studied the impact on noise in the spectrometer system [29] (Denoising analysis of compact CCD-based spectrometer[J]. Optik, 2018, 157: 693-706.), and the effect of noise on the process of noise also has given clear simulation values in the manuscript.

Fig.1 Simulated noise values

N is the total noise, which is the noise value of the current resonance wavelength. It can be seen from the trend in the Fig.1 that as the light intensity decreases, the noise value of the current point can be converged.

We have made a lot of changes to the error you raised. We are benefit a lot from your reviews,thank you very much! 

Sincerely,

Guo Xia

Reviewer 3 Report

This manuscript provides a method to improve the practicality of the plasma sensor by studying the spectral signal-to-noise ratio of the sensor’s general factors. The results indicate that the resolution can be significantly improved (nearly doubled) by increasing the spectral signal-to-noise ratio, which are well verified through simulation modeling and physical experiments. Overall, the results presented in this work are interesting, and the manuscript is well-prepared. I would like to recommend its acceptance for publication in “Sensors” after some minor revisions as noted below.

1.Several grammatical and format errors are found in the manuscript, such as: “Equation (12)” on page 9 is suggested to be revised to “Eq. 12”. Check the grammar of Line 14-15. A more careful check should be carried out before re-submission.

2. The quality of Fig. 7 should be improved.

3. For the noise value simulation in the manuscript, the specific noise simulation process should be provided.

Author Response

Manuscript ID: sensors-1063347 Type: Communications

Title: Effect of Spectral Signal-to-Noise Ratio on Resolution Enhancement at Surface Plasmon Resonance

Author: Long Ma, Guo Xia, Shiqun Jin, Lihao Bai, Jiangtao Wang, Qiaoqin Chen, and Xiaobo Cai;

Dear Reviewer:

Thank you for your careful reading of the article and your outstanding suggestions. We have modified the article in the way you suggested, and hope our modification can satisfy you.The main corrections in the manuscript and responses to the reviewer’s comments are as follows:

Responds to the reviewer’s comments:

  1. Response to comment: Several grammatical and format errors are found in the manuscript, such as: “Equation (12)” on page 9 is suggested to be revised to “Eq. 12”. Check the grammar of Line 14-15. A more careful check should be carried out before resubmission.

Response: Thank the reviewers for their suggestions for this article, we can definitely check the manuscript carefully before resubmitting. The article's grammar and format changes are as follows:

  • The "Equation (12)" and "Equation (13)" were revised to "Eq. 12" and "Eq. 13" in accordance with the manuscript format requirements
  • The SPD used in the manuscript is variable, and we have modified the spectral power distribution to italics.
  • 2 shows that the quoted image has also been marked, and the source of the formulas and parameters in the text.
  • This 2.3.2 title was not stated clearly when submitted, and the title has been revised.
  • The picture description of the paper in the manuscript was not numbered correctly when the picture was modified. We further checked the description of the corresponding picture in the manuscript.
  1. Response to comment: The quality of Fig. 7 should be improved.

Response: We are very sorry that our negligence cause the changes in the resolution of the image to lead to the quality of the image to decrease. We revised the resolution of Fig.7 and reloaded it into the manuscript.

  1. Response to comment: For the noise value simulation in the manuscript, the specific noise simulation process should be provided.

Response: In order to facilitate the reproduction of the article, a clear definition is given for the physical quantities used in the simulation. In the manuscript, the current noise level of each noise is given in the data model for SPR curve.

The gold film is widely used in the SPR system, so in the simulation process we use the gold film as the metal layer of the classic Krestchmann structure, and give the parameters used in the simulation. Our research group has studied the impact on noise in the spectrometer system [29] (Denoising analysis of compact CCD-based spectrometer[J]. Optik, 2018, 157: 693-706.), and the effect of noise on the process of noise also has given clear simulation values in the manuscript.

Fig. 1 Simulated noise values

N is the total noise, which is the noise value of the current resonance wavelength. It can be seen from the trend in Fig.1 that as the light intensity decreases, the noise value of the current point can be converged.

We have made a lot of changes to the error you raised. We are benefit a lot from your reviews,thank you very much! 

Sincerely,

Guo Xia

Round 2

Reviewer 2 Report

The authors have made appropriate changes to the manuscript following on from my earlier review comments. A significant outstanding point is that the figures in the manuscript have now changed but the authors have not updated the figure numbering (in both the figure captions and main text). This needs to be corrected.

Author Response

Manuscript ID: sensors-1063347 Type: Communications

Title: Effect of Spectral Signal-to-Noise Ratio on Resolution Enhancement at Surface Plasmon Resonance

Author: Long Ma, Guo Xia, Shiqun Jin, Lihao Bai, Jiangtao Wang, Qiaoqin Chen, and Xiaobo Cai;

Dear aimee an and Reviewer:

         Thanks to the editors and reviewer for their patient review of this manuscript. Sorry that we did not update the figure numbering in time when edited the picture. We have modified the figure numbering in the manuscript (in both the figure captions and main text). The specific changes are as follows:

We hope that you can be satisfied with the manuscript revision.

Sincerely,

Guo Xia
